# Basement Mapping Using Nonlinear Gravity Inversion with Borehole and Seismic Constraints

**Julio Cesar S. O. Lyrio [1] and Yaoguo Li [2,\*]**

[1] Petrobras, Rio de Janeiro 20231-030, Brazil; jlyrio@petrobras.com.br
[2] Center for Gravity, Electrical & Magnetic Studies, Department of Geophysics, Colorado School of Mines, 16th St., Golden, CO 80401, USA
[\*] Correspondence: ygli@mines.edu

**Abstract:** We present an integrated method for mapping the basement structures of sedimentary basins by combining surface gravity data, seismic imaging, and borehole logging information. The core of the method is a nonlinear inversion algorithm for constructing the shape and depth of the basement from surface gravity data. By using the primal-logarithmic barrier method, we impose depth constraints from the borehole information. The basement depth was imaged by seismic interpretation and incorporated into the inversion as a reference model. As a result, the gravity inversion constructs basement structures that are closest to the seismic input while simultaneously satisfying the surface gravity data and borehole information. We used this new methodology to unveil the basement morphology of the Recôncavo Basin, Brazil. Recôncavo is a syn-rift onshore mature basin that exhibits a strong correlation between oil field distribution and tectonic framework. The seismic imaging in the area is ambiguous, and our approach improved the basement definition and highlighted exploration targets in the studied area.

**Keywords:** gravity inversion; basement mapping; geophysical integration

## 1. Introduction

In oil exploration, the seismic method plays the role of the primary geophysical tool because it provides, in general, higher resolving power than other geophysical methods when investigating on the same scale. For instance, the finer details of structural definition and targets can be determined from seismic images. Other methods, such as gravity surveys, however, are often used to provide complementary information to assist seismic interpretation. For example, qualitative gravity analysis is used in regional studies to identify major structural trends, whereas quantitative techniques, such as gravity inversion, can be used to assist seismic depth migration in salt imaging (e.g., [1–3]). Because of its valuable contribution, the use of the quantitative analysis of gravity data, especially detailed 2D and 3D modeling of complex structures, has significantly increased in recent years. The combination of gravity data and seismic imaging is now common in salt imaging. However, similar efforts seem to be lagging in terms of basement mapping. We hope to contribute to this by integrating gravity inversion with seismic and geologic constraints.

One case in point is the following scenario. Seismic processing and interpretation often produce an image of the subsurface, but the structural image is rarely evaluated against the basic criterion that all available geophysical data should be reproduced through forward modeling. The main reason for the lack of such an evaluation is the prohibitive cost required to perform this for seismic data. However, such evaluation can be carried out for other information, such as gravity data. The benefit of utilizing gravity data is two-fold. First, gravity processing is inexpensive compared to seismic processing, and it can be performed much faster. Second, gravity data provide complementary information about the density distribution in a subsurface, which might potentially improve upon a seismic image in a similar manner, as it helps improve base-salt imaging. We submit that

gravity modeling and inversion may be used as valuable tools to crosscheck and improve seismic interpretation for basement mapping.

The basic premise is that the basement model interpreted from seismic data should be consistent with the known geology and, therefore, should reproduce the gravity anomaly over the same area. If the gravity data produced by the seismic model agree (within the error tolerance) with the measured gravity data, this would have independently verified the validity of the seismic interpretation. On the other hand, a large difference between the predicted and measured gravity data would suggest that the seismic basement image is not entirely valid and needs to be modified. The modification can be guided by structural gravity inversion constrained by available well log information. The changes suggested by the inversion must then come back to the seismic interpretation to refine the previously obtained seismic image. This effectively creates a loop that is completed only when a geological basement model respecting all the available information is generated.

In this paper, we follow the above philosophy and propose an approach that combines the resolving power of the seismic image with the ease of gravity modeling and inversion in mapping basement structures. We assume that a seismic model of the basement relief exists, but it does not agree with surface gravity data. We, therefore, invert the gravity data to construct a modified basement model that is consistent with the seismic result. The central problem is one of estimating the shape and depth of the interface separating two contrasting media by using gravity data. Theoretically, this problem has a unique solution if the density contract is known. In practice, however, this is an ill-posed nonlinear inverse problem, and the solution can be non-unique. The non-uniqueness arises from two distinct sources. The first is the fact that we only know the gravity field at the surface, so many different source distributions in the subsurface can reproduce that field. There will be trade-offs between the density contrast and basin depth. The second reason is the all-present difficulty in applied geophysics that we acquire only a finite number of inaccurate measurements, and there are many models that will reproduce the data within the error tolerance. More information is needed to transform this problem into a well-posed one. Since we are attempting to improve upon a seismically derived basement model, it is logical to use that model as the needed prior information. In addition, we can also use borehole logs as another source of prior information.

There are several approaches to introduce prior information in gravity inversion in order to stabilize the process. For example, Ref. [4] used successive linear approximations to derive a stable solution that is implicitly constrained in shape; Ref. [5] applied low-pass filters to dampen the solution so that a well-behaved basement topography was obtained. Others used a more explicit approach by minimizing an objective function of the model. The advantage of using an explicit model objective function is that it allows for the incorporation of several different types of a priori information by changing the form of the function to be minimized. The authors of [6], for example, minimized the total volume of the causative body. Ref. [7] choose to minimize the moment of inertia with respect to the center of the body or to an axis passing through it. Ref. [8] minimized a function that includes relative and absolute equality constraints in order to introduce smoothness and prior depth-to-interface information. Ref. [9] imposed a smoothness requirement on the vertices of a polyhedron body in salt imaging. Ref. [10] minimized an objective function of density that required the model to be close to a given reference model, and this was smoothed in three spatial directions.

Our method has its principles in the method proposed by [10] but involves absolute constraints and has a model parameterization similar to the method proposed by [8]. The method minimizes an objective function of the model that requires not only the model to be smooth and close to the seismic-derived model, which is used as a reference model, but also to honor well-log constraints. The latter are introduced through the use of logarithmic barrier terms in the objective function (e.g., [11–13]).

We first present our inversion method and illustrate it using synthetic gravity data, simulating a portion of a sedimentary basin. We then apply the method to a set of field gravity data acquired from the Recôncavo Basin, Brazil.

## 2. Methodology

The goal of our inversion is to find a reliable model that approximates the interface separating the sediments and the basement. The interface is assumed to represent the geometry of the basement in a portion of a sedimentary basin. The importance of defining this interface lies in the fact that in some sedimentary basins, especially rift-related ones, the basement geometry controls the distribution of potential oil fields. We restrict ourselves to working with only a portion of a basin since the assumptions involving the physical characteristics of the media, such as constant density, for instance, are more likely to be valid in smaller areas. In addition, this approach seeks to broaden the contribution of the gravity method in oil exploration because it focuses the work at an oil-field scale rather than at a basin scale for study.

To solve the problem numerically, we discretize the basement depth into a set of rectangular patches of a constant size and, therefore, represent the 3D sedimentary basin with a set of contiguous rectangular prisms of a constant density contrast (Figure 1). The tops of the prisms are at the surface, and their thicknesses (or heights) are to be determined from observed gravity data. To allow for flexibility in the model in terms of representing varied basement structures, we required the number of prisms in the model, $M$, to be always greater than the number of gravity observations, $N$. This approach allows for a higher resolution in the recovered models because, in contrast to other inversion methodologies that require the number of observations and prisms to be the same, here, we can have a large number of prisms even when only a small number of field observations are available.

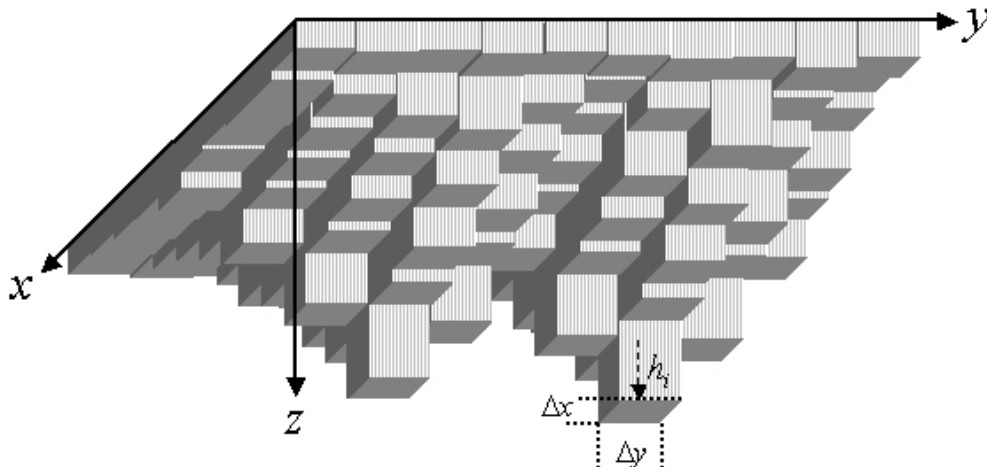

**Figure 1.** Sketch representing the discretization of a sedimentary basin as a set of rectangular prisms having fixed horizontal dimensions $dx$ and $dy$ . The heights of the different prisms, $h_i$, are the parameters to be inverted.

The general relationship between gravity anomaly and its sources is given by (e.g., [14]):

$$g(r) = \int_V \rho(r')\psi(r,r')dv, \tag{1}$$

where $g(r)$ is the gravity field at the observation position $r$ outside of the volume $V$ that is occupied by the source; $\rho(r')$ is the source density at location $r'$, and $\psi(r,r')$ is a function that depends on the geometric relations between positions $r$ and $r'$. Gravity inversion makes use of field measurements to find the main characteristics of either density $\rho$ (linear problem) or some aspects of $\psi$, such as the region of the source. The former is a linear problem, whereas the latter is nonlinear since it intends to recover a geometric aspect of the problem. The problem of recovering the basement depth falls into the latter category.

The relationship between the gravity field at the origin and a single prism with a constant density, $\rho$, and corner positions at $x_i$, $y_j$, and $z_k$, as derived by [15] is,

$$g = \gamma\rho \sum_{i=1}^{2} \sum_{j=1}^{2} \sum_{k=1}^{2} (-1)^i (-1)^j (-1)^k$$

$$\left[ z_k \arctan\left( \frac{x_i y_j}{z_k R_{ijk}} \right) - x_i \log(R_{ijk} + y_j) - y_j \log(R_{ijk} + x_i) \right], \tag{2}$$

where $\gamma$ is the gravitational constant, and $R_{ijk} = \sqrt{x_i^2 + y_j^2 + z_k^2}$. We note that the vertical co-ordinate of the bottom of the prism, $z_2$, is the unknown quantity to be recovered through our inversion.

As discussed in the preceding section, this inversion is ill-posed because we have only a finite number of inaccurate data on the surface, and we attempt to recover a basement relief that is more complex in structure than the smoothly varying gravity data. Consequently, there are a multitude of models that can fit the data to the same degree. In order to find a unique solution for interpretational purposes, we select one that is consistent with known information and is structurally simple. We choose to follow the Tikhonov regularization. This approach allows for the construction of different models by changing the form of the objective function according to prior information. We minimize a total objective function $\Phi$, defined as a weighted sum of a model objective function $\Phi_m$ and a data misfit function $\Phi_d$,

$$\Phi = \Phi_d + \mu\Phi_m, \tag{3}$$

where $\mu$ is the regularization parameter, and it determines the trade-off between the two terms. The data misfit function $\Phi_d$ is defined to be:

$$\Phi_d = \| W_d(g - g^o) \|^2, \tag{4}$$

where $W_d = diag\{1/\sigma_1, \ldots, 1/\sigma_N\}$, in which $\sigma_i$ is the error standard deviation related to the $i$th observation, $g^o$ represents the observed data, and $g$ is the data predicted from the model. If the noise contaminating the data is uncorrelated and has a zero mean, the misfit $\Phi_d$ is a chi-squared variable with $N$ degrees of freedom. The number of observations, $N$, therefore, becomes the target misfit ($\Phi_d^*$) for the inversion since the expected value for a chi-squared distribution is $N$. In the case where the noise statistics are unknown, we must resort to different approaches to determine the optimal data misfit.

The model objective function $\Phi_m$ allows us to incorporate prior information about the model. The choice of prior information is problem-dependent, but in a general sense, the inverted model should be close to a reference model and be as smooth as the data allows in all directions. We, therefore, choose a model objective function having the following form:

$$\Phi_m = \alpha_s \int_S (h - h_0)^2 ds + \int_S \left[ \frac{\partial(h - h_0)}{\partial x} \right]^2 ds + \int_S \left[ \frac{\partial(h - h_0)}{\partial y} \right]^2 ds, \tag{5}$$

where $h$ is the recovered model, $h_0$ is the reference model, and $\alpha_s$ is a coefficient that controls the relative importance of the first term to the others. In Equation (5), the first term provides a measure of the deviation from the reference model, whereas the remaining terms control the structural complexity of the model. Given the discretization used for the forward modeling, the recovered basement depth $h(x, y)$ becomes a piece-wise constant function, and it can be represented by a vector $h = (h_1, \ldots, h_M)^T$. When evaluating the integrals in Equation (5) according to the above-described discretization, we obtain a discrete form of the objective function:

$$\Phi_m = \| W_m(h - h_0) \|^2, \tag{6}$$

where $W_m$ is the model weighting matrix.

The choice of the reference model is often left open in many publications since it is highly problem-dependent. In our inversion, however, the goal is to improve upon seismic interpretation by finding modifications using gravity data. We would like to find a model that deviates as little as possible from the seismic model while still fitting the gravity data. It is, therefore, optimal to use the seismic model as the reference model.

Since the unknown model to be recovered is the height of each prism, the relationship between the data and the model is nonlinear, as discussed earlier. Consequently, the misfit of the data in Equation (4) is not a quadratic function. As a result, we have a nonlinear inverse problem, and we choose to solve it iteratively through linearization. We assume that the thickness is $h^{(n)}$ at the $n'$th iteration, and a small perturbation $\delta h$ can be added to improve the data misfit. By expanding the predicted gravity data using a Taylor series expansion in $\delta h$ and ignoring higher order terms yield a linear relationship,

$$g_i\left(h^{n+1}\right) \approx g_i\left(h^{(n)}\right) + \sum_{j=1}^{M} \frac{\partial g_i\left(h^{(n)}\right)}{\partial h_j} \Delta h_j, i = 1, 2, \ldots, N, \tag{7}$$

where $\Delta h_j = h_j^{(n+1)} - h_j^{(n)}$. Equation (7) can be compactly represented in a matrix form as:

$$g^{(n+1)} = g^{(n)} + J\Delta h, \tag{8}$$

where $g^{(n+1)}$ is the $N$—length vector of the predicted data, $\Delta h$ is the $M$—length vector of model perturbations, and $J$ is the $N \times M$ sensitivity matrix relating the predicted data to the changes in the model at each iteration according to Equation (2). Substituting Equation (8) into the discretized objective function yields the linearized form:

$$\Phi(\Delta h) = \|W_d\left(g^{(n)} + J\Delta h - g^o\right)\|^2 + \mu\|W_m\left(h^{(n)} + \Delta h - h_0\right)\|^2. \tag{9}$$

Minimizing Equation (9) with respect to the model perturbation yields the desired $\Delta h$, which allows us to update the model and proceed to the next iteration.

In addition to the smoothness and similarity to the seismic model, depth-to-basement information (from boreholes) is also available to constrain the solutions. The use of localized prior information as constraints is not new, and examples can be found in [8,16,17], among others. There are different means to introduce localized information, and the majority of methods rely on slightly different ways of minimizing the differences between the estimates and the known depths at well locations. In this paper, we have chosen to apply the logarithm barrier method (e.g., [11]), which has been used by [12,13] in the inversion of different geophysical datasets. One advantage is that this approach allows one to set different limits to every element of the model instead of only at those locations where depth-to-basement information is present. The log barrier method presents the additional advantage of allowing for the introduction of specific degrees of confidence (by narrowing or enlarging the barrier limits) to different information. In other words, it is possible to set very narrow limits at positions where reliable depth information is present and to relax the constraints in regions where the information is less accurate. A fundamental application of these advantages is the introduction of the well log information coming from those boreholes that have not reached the basement but that contain information about depths where the basement certainly is not present. In a very similar way, seismic information can be used in areas where no wells are available. In the log barrier method, such information can be easily incorporated into the inversion by setting the constraints to the minimum depth only. To the best of our knowledge, the use of such information as constraints in the inversion of potential field data is new. The logarithmic barrier method was implemented

in our problem by adding a logarithmic term to the objective function of Equation (9) to form a new objective function:

$$\Phi = \|W_d\left(g^{(n)} + J\Delta h - g^o\right)\|^2 + \mu\|W_m\left(h^{(n)} + \Delta h - h_0\right)\|^2$$
$$-2\lambda\left[\sum_{j=1}^{M}\ln\left(\frac{h_j - a_j}{b_j - a_j}\right) + \sum_{j=1}^{M}\ln\left(\frac{b_j - h_j}{b_j - a_j}\right)\right], \tag{10}$$

where the last term is the barrier function, $\lambda$ is the barrier parameter, $a_j$ and $b_j$ are, respectively, the minimum and maximum depth, and $M$ is the total number of prisms in the model. The barrier term forms a barrier at the boundary of the feasible interval of the unknowns and prevents the minimization from producing unknowns outside their respective bounds. The value of $\lambda$ is decreased during the minimization so that at the end, as $\lambda$ approaches zero, the solution to Equation (10) approaches that of the original problem. Carrying out the complete minimization of Equation (10) for each value of $\lambda$ is an expensive process, and it is also unnecessary. Instead, for each value of the barrier parameter $\lambda$, we take one Newton step towards minimizing Equation (10) to yield the model perturbation equation:

$$\left(J^T W_d^T W_d J + \mu W_m^T W_m + \lambda X^{-2} + \lambda Y^{-2}\right)\Delta h =$$
$$J^T W_d^T W_d(g^o - g) + \mu W_m^T W_m(h_0 - h) + \lambda\left(X^{-1} - Y^{-1}\right)e, \tag{11}$$

where $X = diag\{h_1 - a_1, \ldots, h_m - a_M\}$, $Y = diag\{b_j - h_1, \ldots, b_m - h_M\}$, and $e = (1, \ldots, 1)^T$. The matrix system in Equation (11) is solved for $\Delta h$ by using the conjugate gradient (CG) method. The model is then updated by a limited step-length:

$$h^{(n)} = h^{(n-1)} + \eta\beta\Delta h, \tag{12}$$

where $\beta$ is the maximum permissible step length, and $\eta$ is a parameter that limits the step length actually taken. The parameter $\beta$ is given by:

$$\beta = \min\left(\min_{\Delta h_j > 0}\frac{b - h_j^{(n-1)}}{\Delta h_j}, \min_{\Delta h_j < 0}\frac{h_j^{(n-1)} - a}{|\Delta h_j|}\right). \tag{13}$$

The maximum step length is the value that will take the updated model to the bounds. Limiting it by the $\eta$ prescribed within the interval (0, 1) ensures that the updated model remains within the bounds. After each iteration, the value of $\lambda$ is reduced by:

$$\lambda^{n+1} = [1 - \min(\beta, \eta)]\lambda^n, \tag{14}$$

so that the barrier term becomes negligible as we move towards the final solution. The iterative process is terminated once the barrier term has become negligibly small and the original objective function has reached a plateau. This yields one solution for a given regularization parameter $\mu$. The solutions for several values of $\mu$ are required to find the solution that produces the target misfit $\Phi_d^*$.

### 3. Synthetic Example

We now apply our method to the synthetic dataset shown in Figure 2. The data simulates the gravity response of the model (Figure 2a) at 100 random locations (crosses). Gaussian noise with a zero mean and a standard deviation of 0.04 mGal was added to the entire set of synthetic measurements, resulting in the gravity response shown in Figure 2b. The synthetic gravity data were gridded using 500 m intervals for the purpose of display only.

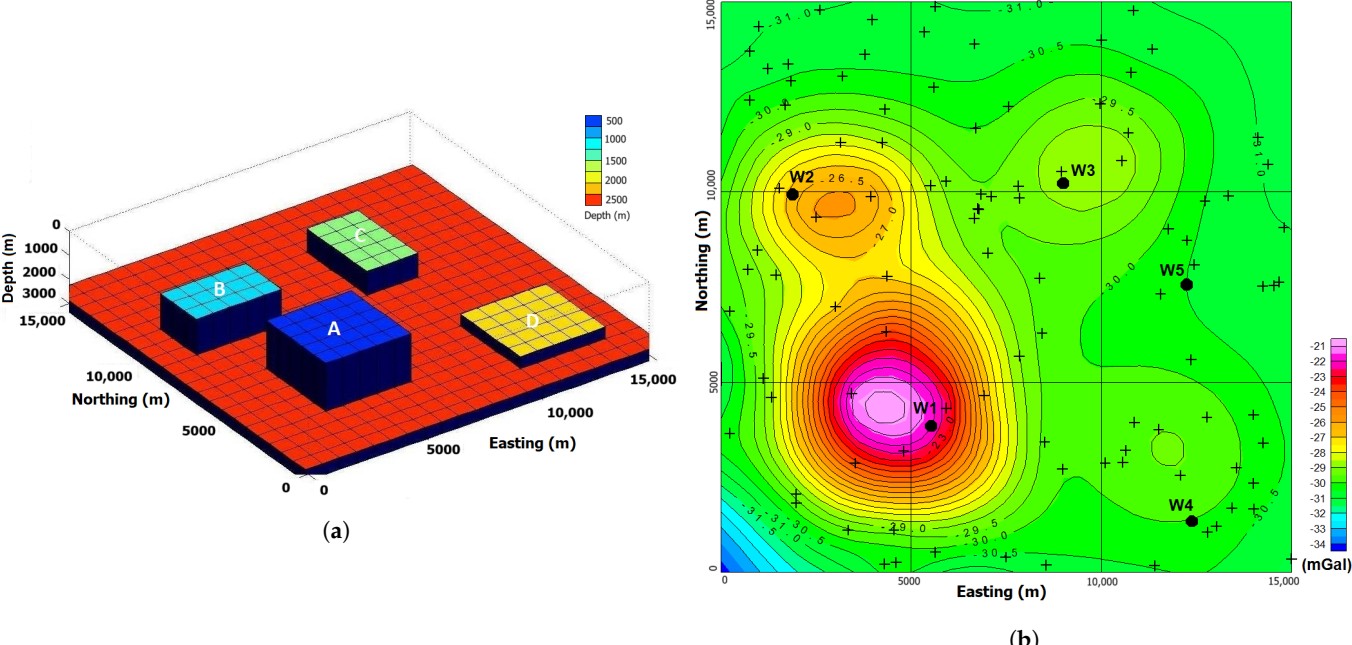

(**a**)

(**b**)

**Figure 2.** Synthetic model representing a restricted portion of a sedimentary basin and its gravity response. (**a**) The model is composed of rectangular features, marked A, B, C, and D, that intend to simulate four structural highs, for which the tops are positioned, respectively, at 500, 1000, 1500, and 2000 m. (**b**) The gravity response of the synthetic model is shown in (**a**), calculated for 100 randomly distributed stations (crosses) by using a density contrast of $-0.30$ g/cm$^3$. The black circles show the position of the five synthetic wells listed in Table 1. The data were gridded with 500 m intervals for the purpose of display only.

The synthetic model shown in Figure 2a simulates a small portion of a sedimentary basin covering an area of 15,000 m × 15,000 m. The basement structures are represented by four rectangular blocks (A, B, C, and D), for which the tops are at, respectively, 500, 1000, 1500, and 2000 m. The maximum depth in the model is 3000 m. The density contrast between the sediments and the basement is considered to be constant and equal to $-0.30$ g/cm$^3$. The model is discretized into 441 rectangular prisms, having a width of 750 m in *x*- and *y*-directions. Since the prisms represent the sedimentary section, the top of each prism is fixed at the surface, and its bottom will determine the depth to the basement at each location, as represented in Figure 1.

The well log constraints were imposed on the problem by assuming the depth-to-basement information at five locations (the black dots in Figure 2b), as listed in Table 1. The wells were incorporated into the model by setting the model's cells at the well locations to provide depth information and keep them fixed during the inversion. Except for the five positions where the depth to the basement is known, a model with a constant depth of 1500 m was chosen as the reference model.

**Table 1.** Location of the five synthetic wells used to constrain the inversion.

| Wells | East Coord. (m) | North Coord. (m) | Depth (m) |
|---|---|---|---|
| 1 | 5522 | 3849 | 500 |
| 2 | 1872 | 9943 | 1000 |
| 3 | 8987 | 10,232 | 1500 |
| 4 | 12,371 | 1345 | 2000 |
| 5 | 12,236 | 7562 | 2500 |

The final result of the inversion is shown in Figure 3a. It is clear that the inversion was not able to completely recover the model, but the results represent a satisfactory solution in terms of the location and average depth for all four structures. The histogram of the absolute data misfit in Figure 3c shows that 83% of misfits are smaller than 0.12 mGal, with 23% below 0.04 mGal, which is the standard deviation of the added noise. Such a result was expected mainly due to the noise and the limited number of observation points. As a comparison, Figure 3b shows the results of a new inversion that used 250 randomly spaced data points. In Figure 3d, the histogram of the absolute data misfit of the new inversion shows that all the misfits are below 0.12 mGal, with 83% of them below the standard deviation of the noise. The increase in the amount of observed data allows for a better definition of the gravity field by reducing ambiguity and helping to improve the final model.

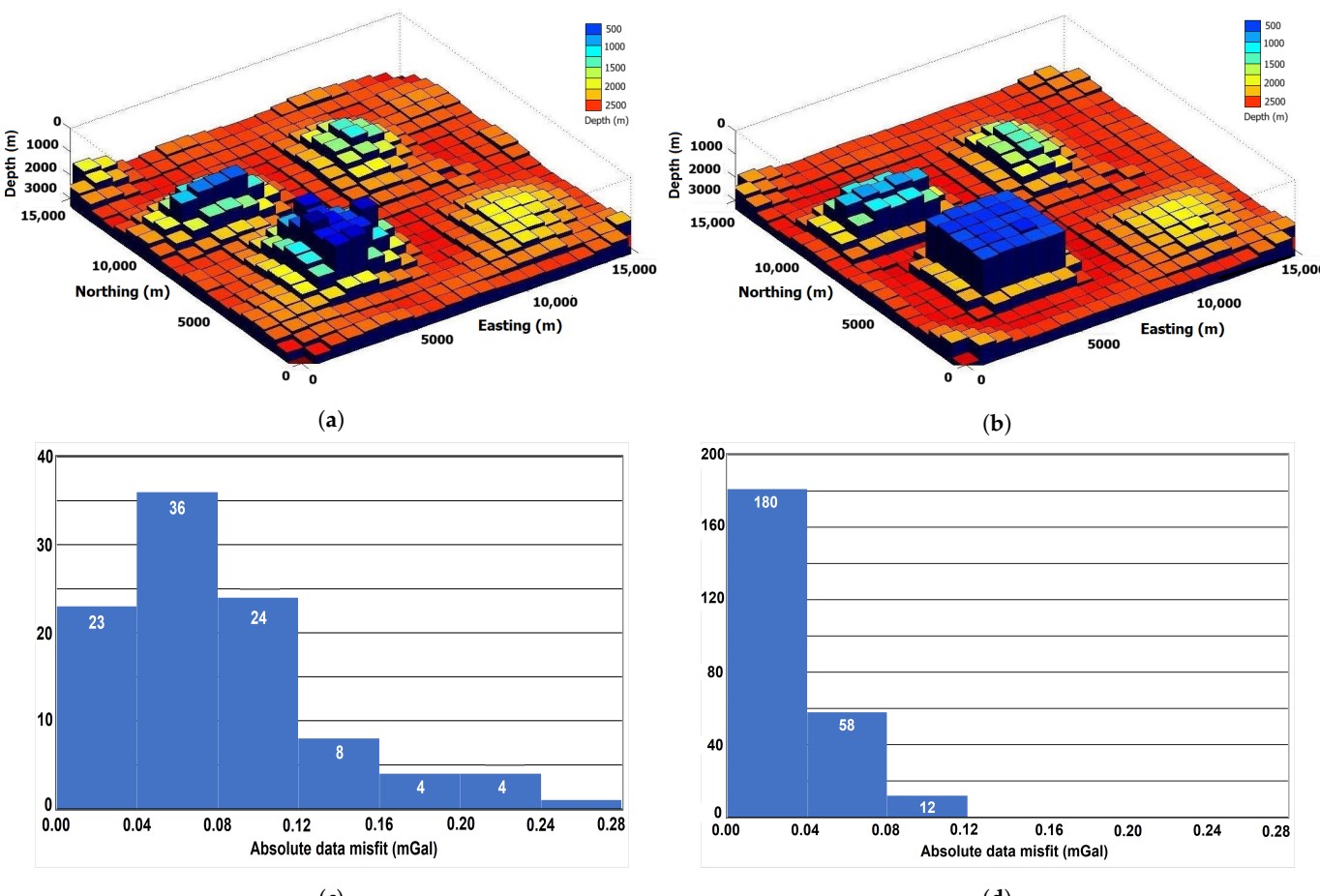

**Figure 3.** Inversion resulting models and respective data misfit histograms. (**a**) Inversion results for 100 noisy contaminated randomly distributed points showing reasonable estimates for locations and average depths for all structures. (**b**) Inversion results for 250 randomly distributed points, which allows better block definition. The histogram of the absolute data misfit for the inversion using 100 points is shown in (**c**), whereas (**d**) exhibits the histogram of the absolute data misfit for the 250 points inversion.

In inversion methods, the correct choice of parameters is usually problem-dependent, and there is no simple rule of thumb available. Therefore, in addition to showing the effectiveness of the proposed method, we also provide the reader with a short discussion on the effects of some of the parameters involved in this inversion process: the parameter $\eta$, the logarithm-barrier parameter ($\lambda$), and the regularization parameter ($\mu$). Based on our experience, we hope that such a discussion can help the readers to develop a feeling for how to choose these parameters for their own problems.

The tests that used different values of the $\eta$-parameter showed that the influence of this parameter on the improvement of the solution is minor, and it is mainly restricted to the speed of convergence. Within the theoretically valid range, the number of iterations increases as $\eta$ approaches zero since the actual step taken at each iteration is too small. As $\beta$ approaches unity, the solution of Equation (11) becomes much more difficult. This is because of the disparity in the elements of matrices $X$ and $Y$, which causes the matrix system to be poorly conditioned. Our tests indicate that values of $\eta$ ranging from 0.9 to nearly 1.0 lead to similar convergence rates and computational costs. For the final solutions shown in Figure 3a,b, the $\eta$-parameter was chosen to be 0.99.

For the logarithm-barrier parameter, we usually start with a large value that must be reduced after each iteration (e.g., [11]). The tests that used different values for $\lambda$ showed that, as expected, the initial choice of $\lambda$ does not produce significant changes in the final solution, and this does not change the effectiveness of the depth constraints. We have chosen the approach in [12] to calculate the starting value of $\lambda$ as:

$$\lambda = \frac{\Phi_d + \mu\Phi_m}{-2\sum_{j=1}^{M}\left[\ln\left(\frac{h_j - a_j}{b_j - a_j}\right) + \ln\left(\frac{b_j - h_j}{b_j - a_j}\right)\right]}. \tag{15}$$

The choice of the regularization parameter $\mu$ is the most important step towards a good inversion result. It should be noted that $\lambda$ is an auxiliary parameter that does not directly change the final results, whereas $\mu$ is the parameter that determines the trade-off between model complexity and data misfit. Therefore, the parameter $\mu$ directly affects the final result, and its choice is crucial. The $\mu$ parameter is often chosen so that the misfit term reaches the target misfit at the final iteration. Such a criterion works well for cases where the noise is uncorrelated and zero-mean, and a good estimate of the standard deviation of this noise is available, as in the synthetic example presented here. Unfortunately, such cases are rare in practical applications.

When no information about data errors is available, other methods for estimating the regularization parameter must be used. [18] suggested the use of either GCV or L-curve criteria as an effective automatic estimator of the trade-off parameter in nonlinear inverse problems. We have found that the L-curve criterion produces good $\mu$ estimates for the synthetic examples in our problem. Therefore, we have incorporated this criterion in our inversion methodology by using the maximum curvature approach proposed by [19] to automatically locate the L-curve corner.

## 4. Recôncavo Basin Example

### 4.1. Geologic Regional Settings

As an example of application to a real problem, we have applied the proposed method to estimate the relief of the basement in a small portion of the Recôncavo Basin (RB), Brazil (Figure 4). Located in the country's northeast region, the mature RB is the oldest oil province in Brazil [20]. The RB is an aborted branch of a larger rift complex called the Recôncavo-Tucano-Jatobá rift, formed by a series of elongated half-grabens separated by oblique transfer faults extending over 620 km across the continent [20].

The origin and evolution of the rift are related to stresses that occurred in Gondwanaland during Mesozoic times, before continental drift [21]. The RB is characterized by a strong correlation between the distribution of oil fields and basement structures, which makes the correct understanding of the basement framework fundamental.

The stratigraphy of this basin can be simplified into two main sequences: the pre-rift and rift sequences [20]. The pre-rift sequence lays directly above the basement and is characterized by thick packages of sandstones, which are the principal reservoirs in the basin. Overlaying the pre-rift sequence is the rift sequence. This sequence is characterized mainly by shales, including the area's source rock. The thickness of the pre-rift sequence is almost constant along the entire work area, whereas the rift sequence is thicker in the

structural lows of the basin. Two major fault systems, trending NE-SW and NW-SE, are responsible for the structural complexity of the area. Most faults directly connect reservoirs and source rocks, forming most oil fields over the internal structural highs. Figure 4 shows a simplified stratigraphic section in the work area.

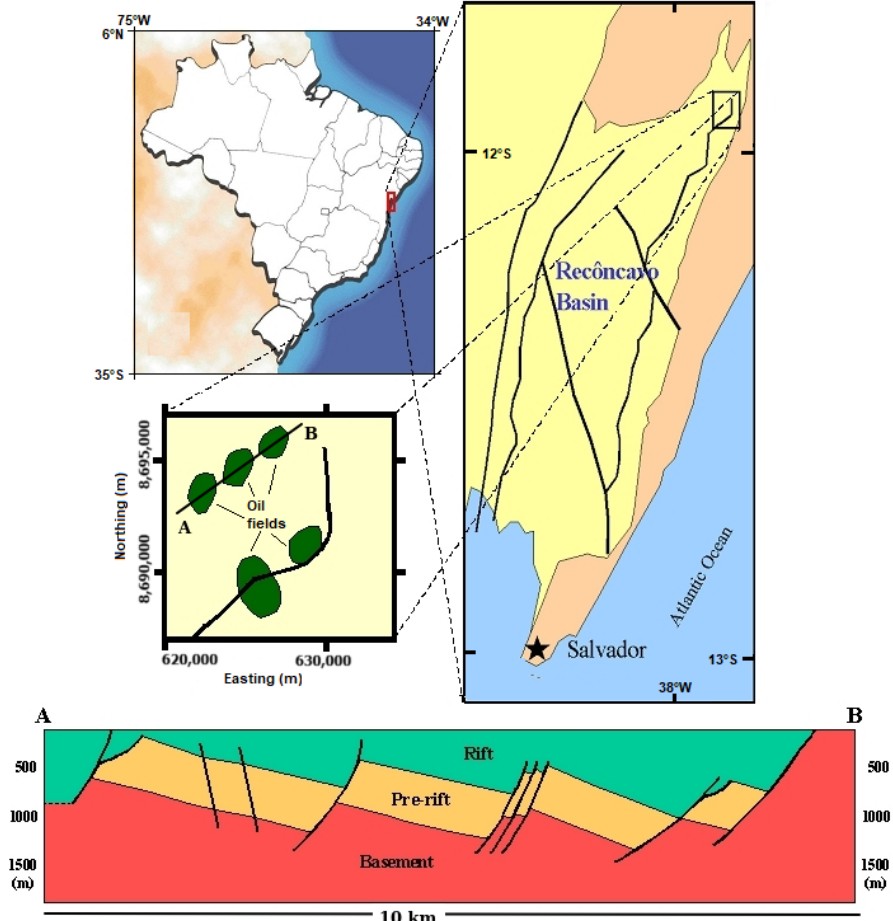

**Figure 4.** The geographic location of the Recôncavo Basin in Brazil. The detail shows the area chosen for the field example and the position of the schematic cross-section AB. The section shows the structural relationship between the three main stratigraphic features present in the work area.

### 4.2. Multiphysics Dataset

The studied area has been intensively explored, with several geophysical surveys acquired over several decades. The Brazilian National Petroleum Agency (ANP) provides free access to the public gravity and 2D seismic datasets used for the present interpretation.

The gravity data herein are interpreted as a subset of the onshore Debardenest regional gravity dataset made available by ANP. All land gravity stations were tied to the 1971 International Gravity Standardization Network (IGSN-1971) and were processed through a standard workflow to recover the Bouguer anomaly [22]. Our subset comprises 771 stations that are nearly uniformly distributed in a grid over the studied area, as shown by the crosses in the Bouguer map of Figure 5a. The data were interpolated to a regular grid using 250 m intervals.

The map in Figure 5b is the residual gravity anomaly that was used as the observed data for the inversion. These data were computed by removing a linear trend from the Bouguer anomaly map of Figure 5a. Although regional-residual procedures can change the amplitudes of anomalies in ways that can affect the depth estimates, this additional processing was required before the inversion for basement relief to remove the gravity effect of crustal thinning observed along the sedimentary basins of the northeast Brazilian continental margin (e.g., [17,23,24]).

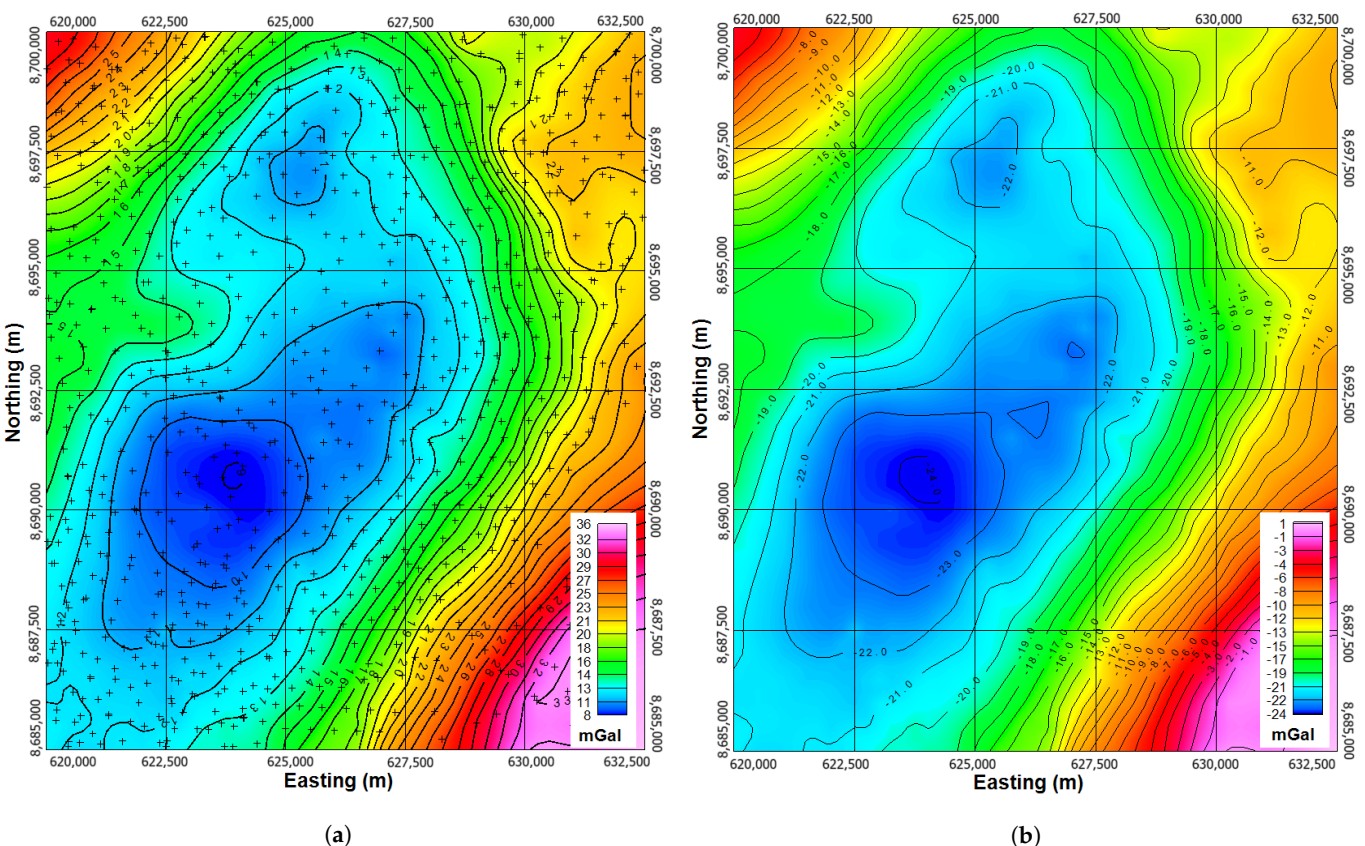

(**a**)    (**b**)

**Figure 5.** (**a**) Bouguer anomaly map of the work area showing the position of the 771 gravity stations (crosses) where the measurements were made. (**b**) Residual anomaly map computed by removing a linear trend from the Bouguer anomaly. This residual is the observed data for the inversion. The data were gridded using 250 m intervals in both figures.

Despite the dense seismic coverage in this part of the basin, as shown in Figure 6a, no seismic basement map is available due to the poor quality of the seismic data. In a paper on the seismic characterization in Recôncavo Basin, [25] states that the large thickness of the recent sedimentary coverage associated with intense cultural activity in many areas makes it difficult to define the position of deep targets (Figure 6b). Besides, energy transmission difficulties and interfingering stratigraphy degrade the seismic signal in the studied area, affecting the signal-to-noise ratio and making the reconnaissance of basement reflections ambiguous. Because of that, reasonable basement estimates have been made from the top of the pre-rift sequence. The pre-rift sequence has two important characteristics in the study area: it is a well-defined seismic reflection that can be easily mapped and has a nearly constant thickness, averaging around 400 m. Due to these characteristics, it has been a typical and thriving practice in this portion of the basin to estimate the basement depths by adding 400 m to the top of the pre-rift sequence mapped from seismic data. We used this practice to get the basement estimate, as shown in Figure 7, which we used as a reference model.

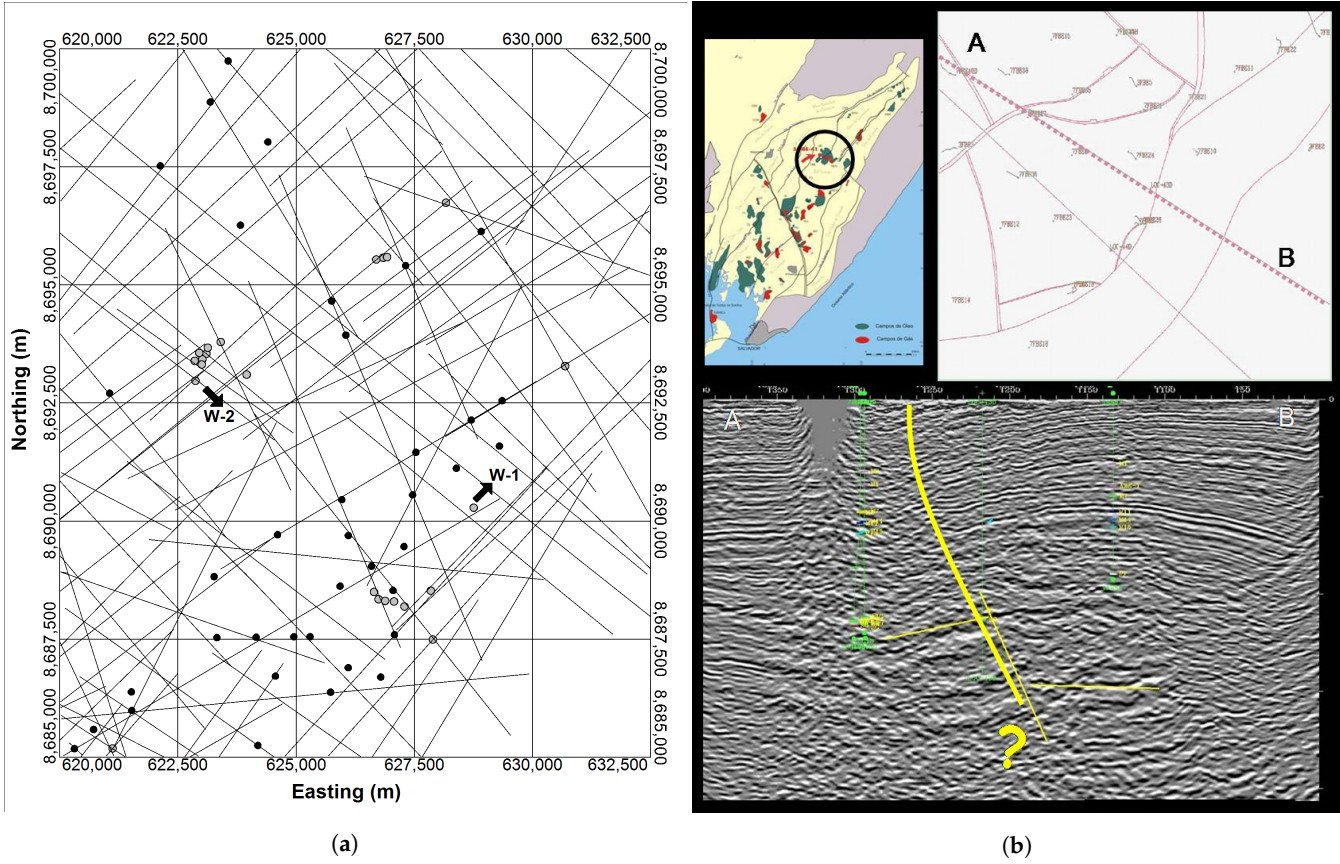

**Figure 6.** (**a**) Position of seismic lines and wells used as constraints. The line segments are the 2D seismic lines used in the mapping of the top of the pre-rift sequence. The gray dots represent the location of the wells that provide reliable depth-to-basement information, and the black dots show the wells that provide constraints on the minimum basement depth only. (**b**) Example of the difficulties in defining the position of deep targets (reflections near the yellow question mark) caused by energy transmission problems, cultural interference, and complex stratigraphy, which degrade the seismic signal. The top left panel shows the location of this example in the Recôncavo Basin, whereas the top right panel shows the direction of the seismic section AB shown in the bottom panel (modified from [25]).

The depth constraints for this inversion come from 61 wells that are distributed throughout the area (Figure 6a). From this total, 23 wells provided direct depth-to-basement information (gray circles). The remaining 38 wells stopped inside the sedimentary section (black circles), providing only lower bounds to the basement depths at those locations. Although density logs were available for some of these wells, only the depth information was used to constrain the inversion.

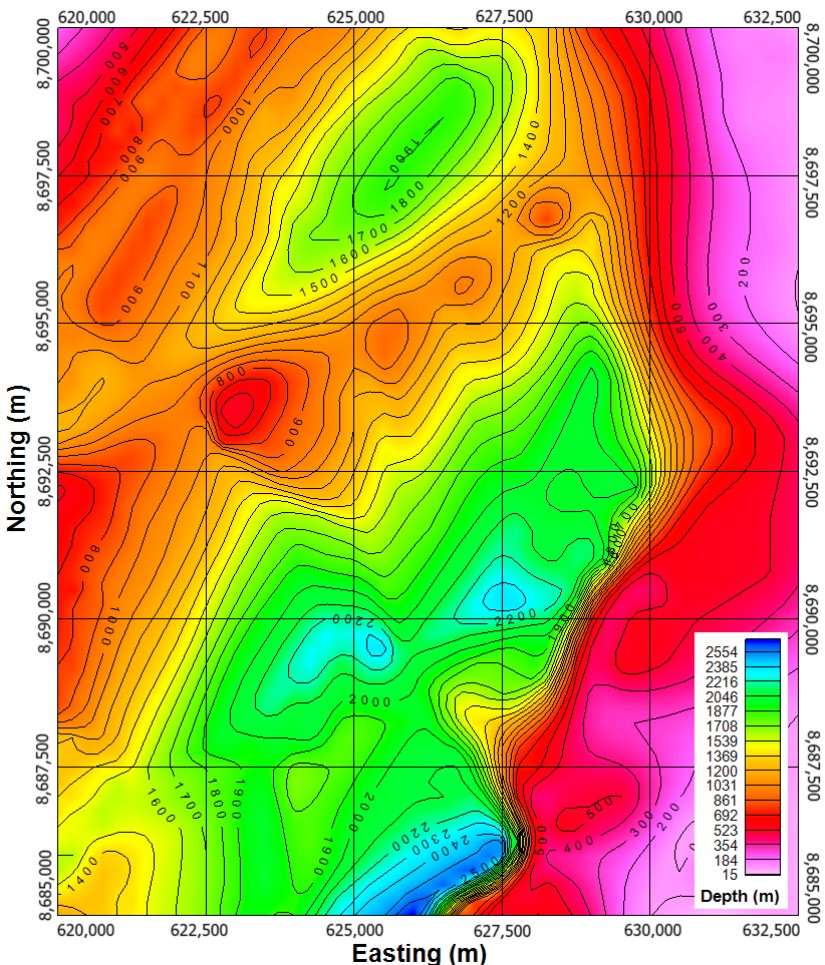

**Figure 7.** Structural map of the basement, as derived from the seismic mapping of the top of the pre-rift sequence (see text for details). These data represent the reference model for the inversion. The data were gridded using 250 m intervals.

*4.3. Basement Relief Estimation*

We assume an average contrast between the basement and sediments. The assumption of constant density contrast is a drawback of the technique since it is an approximation. However, we choose to adopt it because it is the most straightforward approach to be applied when the knowledge about the density distribution in the area needs to be improved, for example, in regions where the exploration is just beginning. In addition, in areas with reduced dimensions, like the study area, both the basement and sediments are not expected to change densities, and an average density contrast is a reasonable approach.

Despite the advanced geological knowledge in the study area, we decided to include only the minimum amount of information required by the inversion methodology. Such an approach allows us to better evaluate the technique's performance against the known geology. We emphasize that, in general, all available information should be used. For example, in cases where the density distribution is known to a certain depth, we suggest using techniques like gravity stripping [26] to remove the effect of known layers and then do the inversion for lower levels using a constant density contrast. Such a simple approach will reduce the complexity of the models, saving computer power and time during the inversion.

The average density contrast used during the inversion was estimated by considering the gravity response of the seismically derived model for different density contrasts at the exact position of those wells whose depth-to-basement is known. Since the seismically derived model honors the basement depths at the well locations, it is reasonable to assume

that the most suitable average density contrast should be the one whose gravity response gives the smallest RMS error compared to the observed gravity at the well locations. According to this approach, the most appropriate density contrast is $-0.39$ g/cm$^3$. This average value is reasonable since it is within the range of density contrasts measured in several density logs distributed over the area.

The gravity response calculated for the seismically derived model using the estimated density contrast is shown in Figure 8a. The predicted data of Figure 8a should be similar to the observed data of Figure 5b if the seismically derived model were correct. There is a good resemblance between the two maps regarding shape, but the predicted data is a smooth version of the observed data. The discrepancies between predicted and observed data are better analyzed in the map of differences in Figure 8b. A quick statistical analysis shows that the amplitudes of the differences, ranging from $-12$ to 6 mGal, are very high. The same can be said about the standard deviation of 3.2 mGal. Analysis of the spatial distribution of the differences points to coherent features in the map, particularly an elongated positive feature, trending SW-NE at the lower center, which suggests an excessively deep basement. If the basement model was correct, the difference map should be dominated by incoherent features mainly related to noise. Therefore, although this model has been considered for a long time as a satisfactory estimate for the basement, it only partially justifies the gravity data and should be re-evaluated. We then applied gravity inversion to modify the basement depths and to reduce the discrepancies between predicted and observed gravity fields.

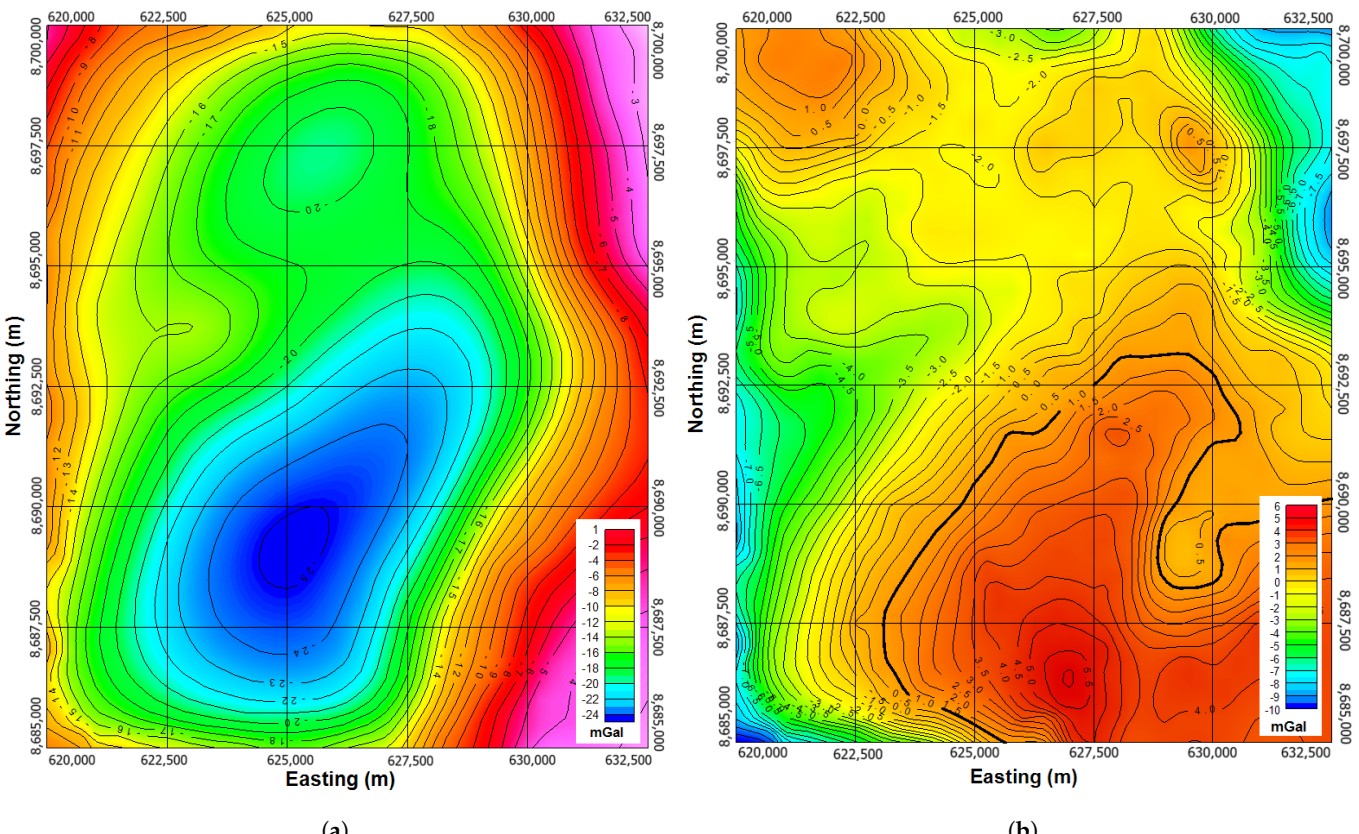

(**a**)            (**b**)

**Figure 8.** (**a**) The residual gravity anomaly map, as calculated from the seismic-derived model of Figure 7 by using a density contrast of $-0.39$ g/cm$^3$. These are the predicted data for the seismic-derived model. (**b**) The differences calculated between the observed and predicted data. Notice that the presence of positive differences (greater than 1 mGal) in the SE corner (thick contour) indicates deficiencies in the seismic model that may be reduced by gravity inversion. The data were gridded using 250 m intervals in both figures.

Following the approaches developed in this paper, we select the inversion parameters as follows: $\gamma$-parameter is set to 0.99, initial $\lambda$-parameter is equal to $10^{-4}$, and $\mu$ is equal to $10^{-10}$. The result of inverting the observed data is shown in Figure 9a. The gravity-inverted basement is very similar in shape to the reference model of Figure 9a. This was already expected since the method requires the maximum possible similarity with the reference model. Figure 9b shows the differences between the reference and the model predicted by gravity inversion. Although the amplitudes of the differences range from −600 to 700 m, the average value is around −70 m. In the majority of the area, however, the amplitudes stay between −200 and 200 m, which indicates a reasonable agreement between the two models. The presence of large values at NE and SW corners can be related to some kind of border problems. Such effects may suggest that the first-order trend used for the regional-residual separation was not a reasonable approximation to the regional field at these locations. The most remarkable feature is the large positive difference situated at the central-south of the map. This feature is the most important contribution to the study area since the inversion has suggested the basement is over 300 m shallower than what was initially estimated from the seismically derived model.

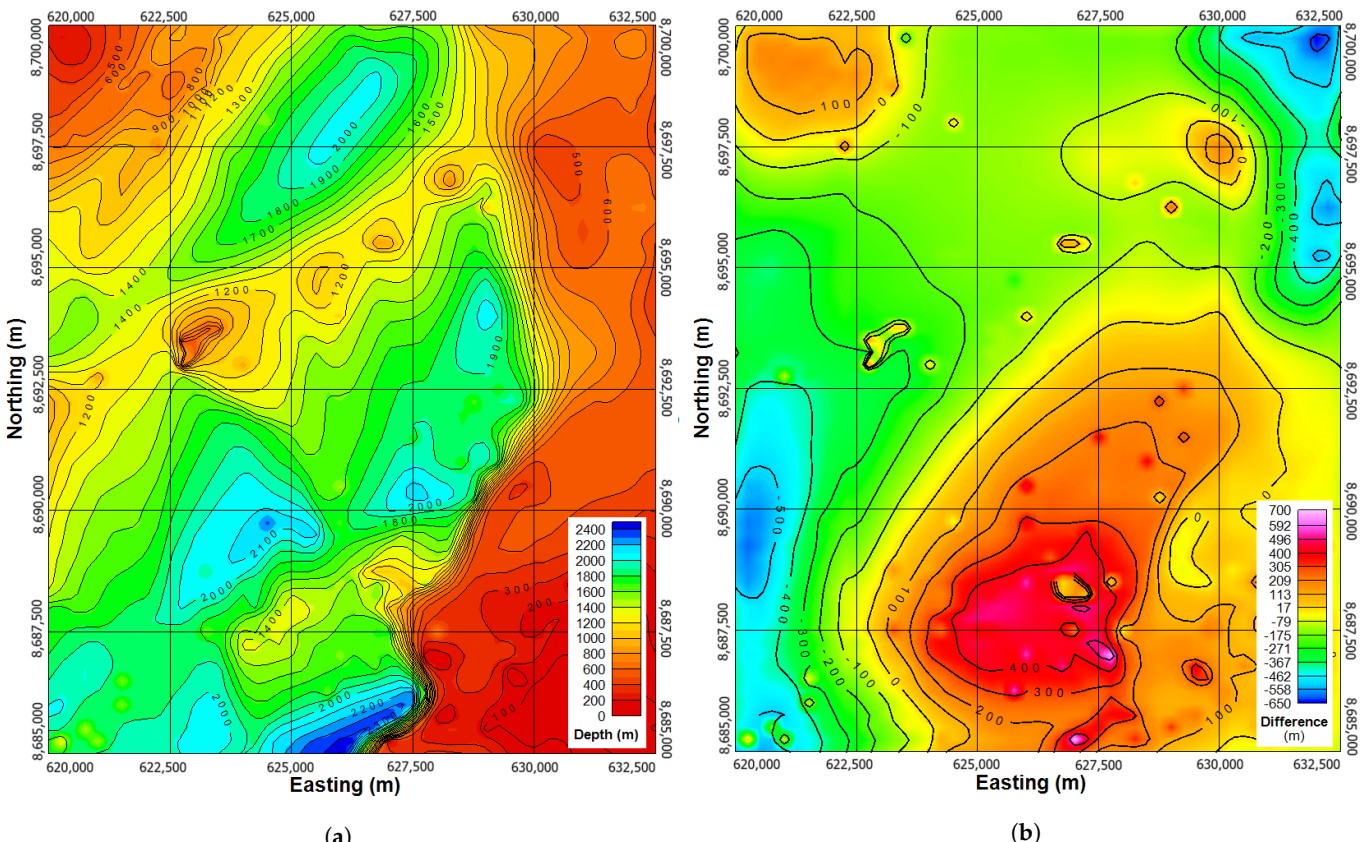

(**a**)　　　　　　　　　　　　　　　(**b**)

**Figure 9.** (**a**) The new structural map of the basement after gravity inversion. Note the overall similarity to the reference model of Figure 7. (**b**) The differences between the seismic-derived and the gravity-inverted basement estimates. A larger difference at the SE corner (over 100 m) correlates with the feature found in the difference map of Figure 8b. The correlation between the two features seems to indicate that the inversion has considered the seismic basement as being too deep to fit the anomaly of Figure 8b and had to raise it in order to fit the anomaly. The data were gridded using 250 m intervals in both figures.

The predicted gravity field resulting from the inverted basement is shown in Figure 10a. In terms of shape, there is a great resemblance between this map and that of Figure 8a, the predicted field from seismic. Such behavior was already expected since only finer modifications were introduced in the seismic-derived model by inversion. The difference,

however, is more significant in magnitude. As an example, the small gravity low toward the north is now deeper, whereas the large low at the south is shallower. The map of differences between observed and predicted gravity, in Figure 10b, also demonstrates the improvement in the fit. The amplitude of the differences now ranges from −10 to 6 mGal, with a standard deviation of 1.5 mGal, which is approximately half of that achieved with the seismically derived model.

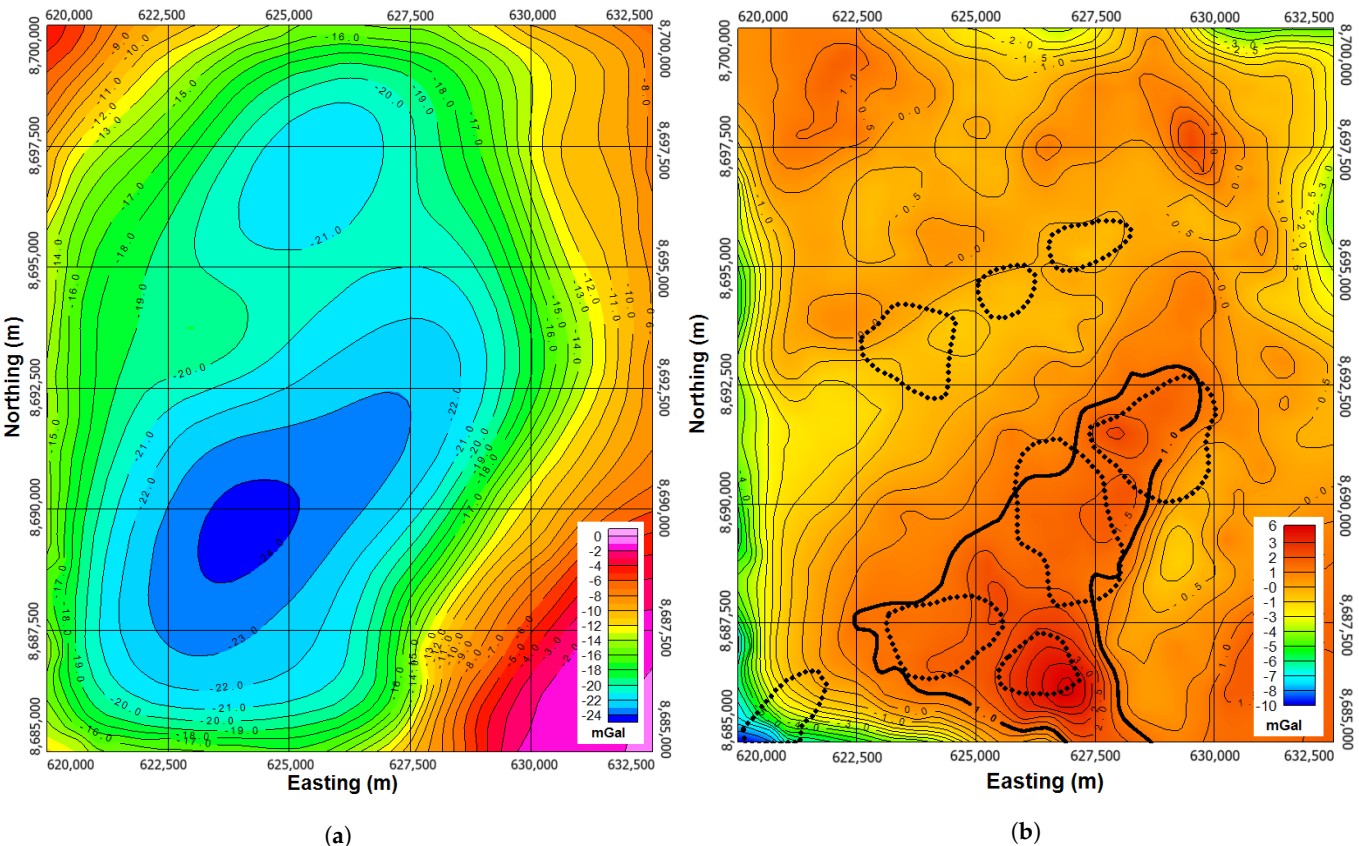

(**a**)                                                                                                    (**b**)

**Figure 10.** (**a**) The predicted gravity field calculated by using inversion. Since only small changes were introduced into the model, the similarity to the field predicted by the seismic model (Figure 8a) is evident. (**b**) The differences between the observed and predicted data. Note the changes in the amplitude of the differences, especially in the two main lows, when compared to Figure 8b. The SW-NE structure (highlighted with a heavy line) in this map is justified by the presence of a trend of oil-bearing high-density sandstones (dotted lines) that are coincident with the structure. The discovery of such anomalous features gives important exploratory significance to this map. The data were gridded using 250 m intervals in both figures.

The improvement is more evident in the histograms of the absolute differences shown in Figure 11. Despite the general reduction in the misfit, the presence of the SW-NE structure in Figure 10b (heavy line), showing differences greater than 1 mGal, indicates that the raising of the basement imposed by inversion was not enough to completely fit the data. It is possible that the presence of constraints has prevented the inversion from raising the basement to justify the anomaly. If the basement is restricted to deeper parts and cannot be responsible for the gravity anomaly, then it is likely that density variations within the sedimentary section are the contributing factor. In fact, such a structure shows a strong correlation with a trend of oil-bearing sandstones (dotted lines in Figure 10b).

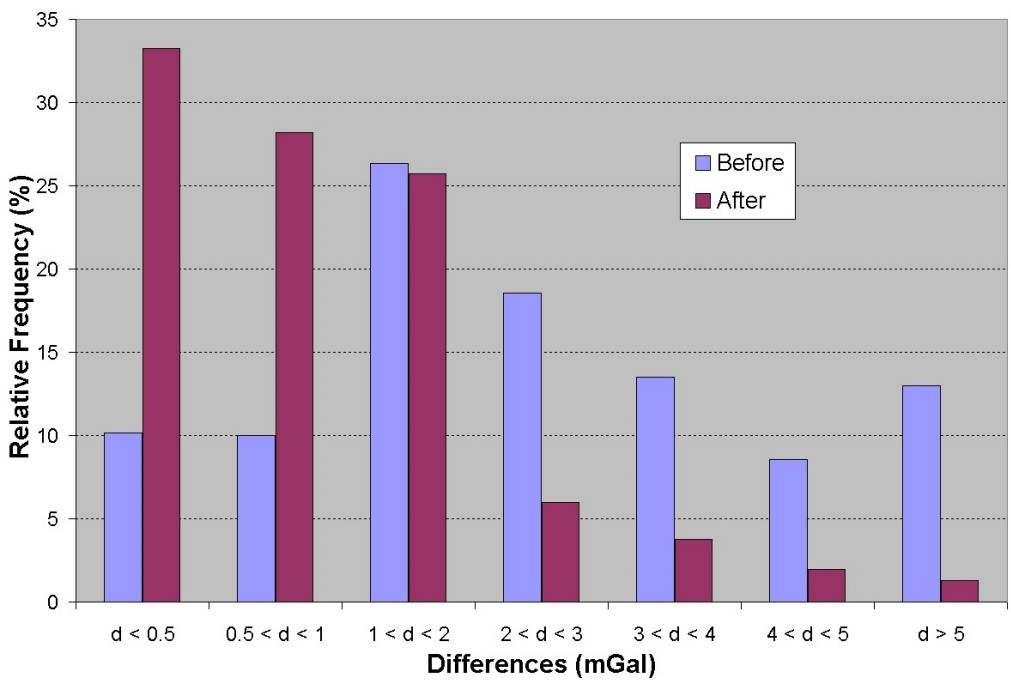

**Figure 11.** Comparison of the differences between the observed and predicted data before and after gravity inversion. The increase in the relative frequency for the absolute differences below 2 mGal after inversion, comparatively to before, proves the fitting improvement.

Detailed investigations about the characteristics of these reservoirs show they are formed by anomalous high-density sandstones located inside the rift section, as proved by the density log of Figure 12a. The region of the log that corresponds to the reservoirs shows densities similar to the basement. In contrast, in the case of a well located out of the trend (Figure 12b), the densities are lower, and there is significant contrast only with the basement at the bottom of the log. Since these intra-sedimentary density variations were not accounted for in the model, it was expected that the inversion would try to fit this anomaly, making the basement shallower in this area. However, the presence of the constraints, especially those limiting the minimum basement depth, avoided such compensation. In this case, therefore, the presence of residuals in the final result points to potential high-density regions correlated to oil-bearing stratigraphic features, which would have important exploratory significance.

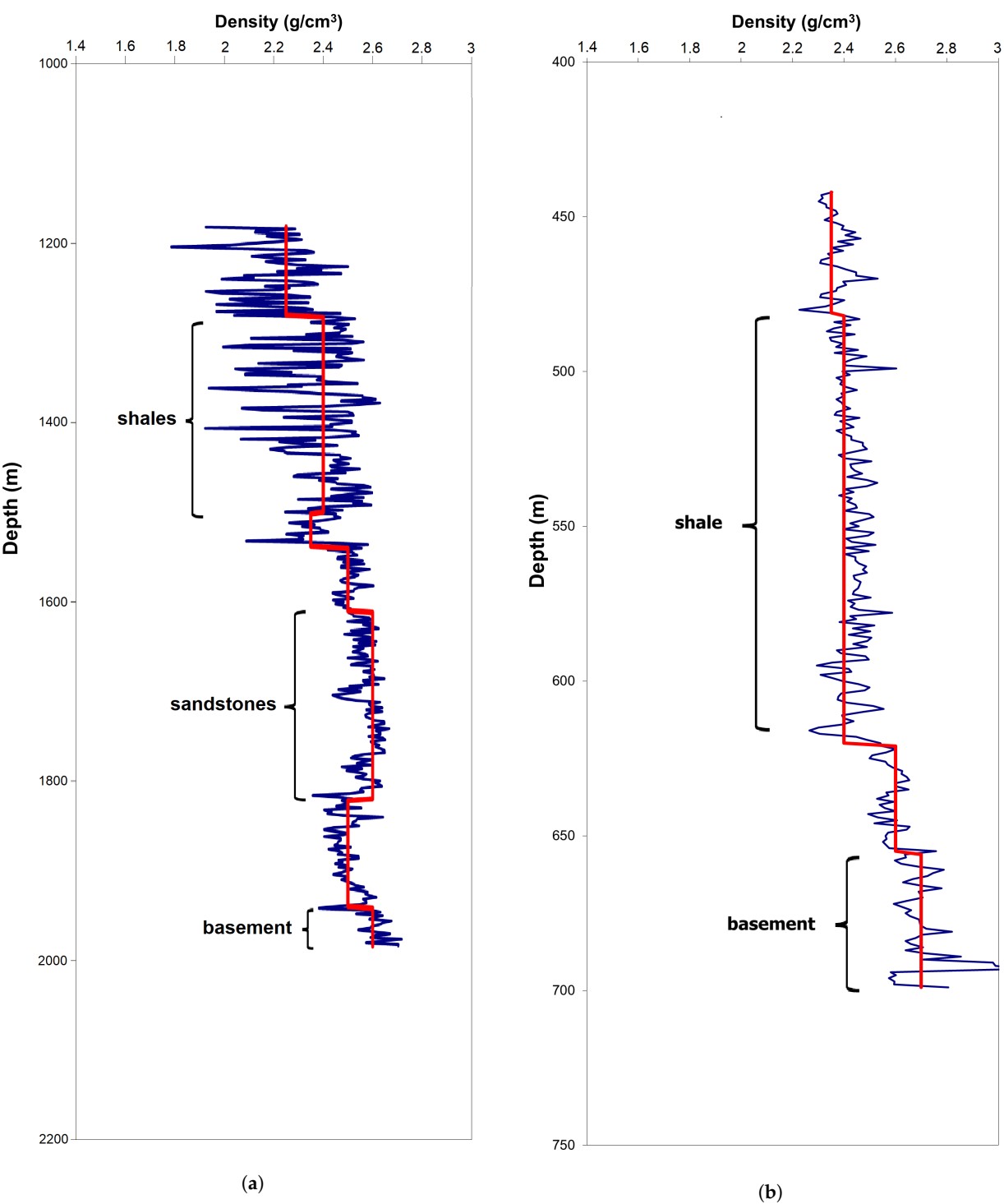

(**a**)  (**b**)

**Figure 12.** Density logs from the two wells at different positions: (**a**) inside the anomaly area (W-1), and (**b**) outside the anomaly area (W-2). Notice that the sandstones in (**a**) show a density similar to that of the basement. The position of the wells is given in Figure 6a. No density data were collected in the shallower portions of both wells.

## 5. Conclusions

We have proposed a new approach for estimating the relief of a surface separating two media of different densities. The method is based on inversion of gravity data that incorporate seismic interpretation and borehole logs. Similar to other inversion methods, the proposed method minimizes the objective function of the model that controls the data

misfit and penalizes deviation from a reference model and the structural complexity of the model. The use of the logarithmic barrier method allows for the incorporation of well log information, even from wells that do not reach the basement. Since such wells are often more common than those penetrating the basement, the number of constraints increases. Furthermore, the logarithm barrier method enables the use of general bounds that vary with location within the basement model. The possibility of including such specific information in the inversion increases our confidence in the final model.

The proposed methodology was successfully tested on a synthetic dataset. The inversion has satisfactorily recovered both the position and the average depth-to-the-top of the structures present in the model. The errors in the recovered model seem to be caused by the fact that the inversion was applied to a limited number of noisy observations. Significantly better results can be achieved either by increasing the number of observations or by reducing the noise level.

The proposed approach was applied to a field gravity dataset acquired in the Recôncavo Basin, Brazil. This basin was chosen because it hosts a number of oil fields located at structural highs that correlate well with gravity anomalies. The dense gravity coverage in the area also makes it ideal for testing the new algorithm. Incorporating the seismic-derived model into the inversion and imposing well constraints on the depth to the basement produced a different basement model that significantly reduced the gravity data misfit. The method has been shown to be effective in the sense that it provides the necessary adjustment to the seismic model in order to produce a satisfactory fit to the observed gravity field.

In addition, the use of a constant density contrast (although for a very simplistic model for sedimentary basins) has proven to be appropriate in the case studied because it was constrained by a large amount of independent information, like seismic data and wells. By preventing the inversion to improve the fit, the presence of constraints led to inversion residuals that were related to density anomalies inside the sedimentary section. The mapping of these density anomalies is of significant importance for exploration in Recôncavo Basin because they are closely related to a specific kind of stratigraphic prospect.

**Author Contributions:** Conceptualization, J.C.S.O.L. and Y.L.; methodology, J.C.S.O.L. and Y.L.; software, J.C.S.O.L.; validation, J.C.S.O.L. and Y.L.; formal analysis, J.C.S.O.L.; investigation, J.C.S.O.L.; writing—original draft preparation, J.C.S.O.L.; writing—review and editing, Y.L.; visualization, J.C.S.O.L.; supervision, Y.L. All authors have read and agreed to the published version of the manuscript.

**Funding:** This research received no external funding.

**Data Availability Statement:** The data used in this research may be requested to the Brazilian National Petroleum Agency-ANP at https://www.gov.br/anp/pt-br/assuntos/exploracao-e-producao-de-oleo-e-gas/dados-tecnicos/legislacao-aplicavel/bndg-banco-nacional-de-dados-gravimetricos, accessed on 24 July 2023.

**Acknowledgments:** The authors thank the Special Issue Editor Paulo T. L. Menezes for the valuable suggestions. Julio Cesar S. O. Lyrio would like to thanks PETROBRAS for permission to publish this work.

**Conflicts of Interest:** The authors declare no conflict of interest.

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
