# Peer review of "Basement Mapping Using Nonlinear Gravity Inversion with Borehole and Seismic Constraints"

_minerals, doi:10.3390/min13091173_

Round 1

Reviewer 1 Report (Previous Reviewer 1)

Unfortunately most of the changes that were previously requested (twice) have still not been made. I note that other reviewers are complaining about the same problem as well. I am trying to assist the authors by providing constructive comments that will improve their manuscript and bring it up to a publishable standard, but this is proving hard work. Note that a review process is not some kind of a vote between the reviewers’ requests as the authors seem to suggest in their response. If one reviewer finds a fault or error that other reviewers have missed, that does not mean that it can be ignored. Different reviewers often have different skillsets and experience. In summary, it is not necessary that all reviewers request a particular correction before the authors need to implement it, as the authors have chosen to decide, i.e. all comments from all reviewers should be implemented, not just a small subset that the authors have selected. Obviously if reviewers make contradictory requests then the journals’ AE can handle the situation (in my experience this is not common). Reviewing a manuscript can be several hours of hard work, and it is frustrating when authors just ignore the comments.

Previously I asked ‘Since borehole data is available, was the density of the basin actually constant with depth ? (ie please plot Figure 12 with depths from 0-2200m).’, ‘… in the particular real data case that is presented in this manuscript, was the density constant with depth ??’ The authors response is that the density data was not collected in the shallower portion of the borehole. That’s fine, but please add a comment in the caption of figure 12 about this.

Previously I stated ‘p.3; ‘This approach allows higher resolution…’; higher resolution than what ?’. Unfortunately the authors have not clarified what they mean in the text of the manuscript. This minor edit would take the authors a few seconds to make, but again ‘we didn’t include it into the text because no other reviewer has required for additional explanation about this subject’. See my introductory comments.

Figure 5; as requested (twice) previously, display the data as an image, not a contour map with a large contour interval. See also Figures 7,8b, 9a, 10b. The authors response is that they do not understand the question. Figure 5a uses a large contour interval of 2 mGals, which results in only about 18 different colours. The map therefore has large flat regions where different gravity values are displayed in the same colour. If displayed as an image then it would have 256 different colours, enabling subtle detail to be seen. In Matlab here is the code

% Contour map

>>contourf(data,10); axis equal tight xy; % Contour map with 10 levels

% Rather use this instead

>> imagesc(data); axis equal tight xy; colorbar;

>> colourmap(jet(256)); % Image with 256 different colours

Other software can also be used of course.

As pointed out previously, Figure 6 is four plots but the caption only mentions a) and b). Make sure that they all have axes labels and units (see also Figure 4). Note that even if the figure was scanned from another journal it can still have labels etc added in Photoshop. This is not much work. Adding axes labels does not ‘change the figure drastically’ as the authors suggest, and the figure is already described in the caption as ‘modified from [25]’.

Previously I requested ‘Could the authors please include information as to the computational cost of the proposed method, including aspects such as L-curve determination etc (ie runtimes, hardware, memory usage etc) ?’ and they authors indicate that because no other reviewers have requested this they are not prepared to do it. Please see my introductory comments.

Previously I requested ‘p.9; show the L-curve’; the authors response is that their code does not display the L curve, it just calculates it in the background and picks the optimum value automatically. Relying on software to automatically choose any important model parameters without any possible method of verification is an extremely bad way to do anything, not just geophysics. Please show an example L curve.

As pointed out previously (twice now), the paper is very short and contains only a single synthetic data example and a single real data example. It would benefit from more of both. I suggest also using a 2D synthetic model initially because the results are easier to visualize, and of course the method is essentially identical. Please see my introductory comments.

As commented previously, unfortunately there are no comparisons with other interpretation methods so the reader does not know whether the proposed method performs better than standard methods such as Li and Oldenburg etc. This needs to be done. The authors response is that they are presenting an approach that works and that that is sufficient. It is not, particularly if the proposed approach has significant disadvantages compared to existing methods (eg it is computationally very slow, or it is less accurate, etc). See my introductory comments.

As commented previously, ‘Figure 3; show the forward response of the models and the difference between it and the (synthetic) observed data.’. I understand that the model is being evaluated only at certain points, but if the model response does not even fit the data then the model will not be valid. See my introductory comments.

Many changes are needed to bring this manuscript up to a publishable standard, but when they are completed I believe that the manuscript will be much improved, and that it will make a positive contribution to the literature.

Author Response

Reviewer 2 Report (Previous Reviewer 2)

I have no commentes any more. Thanks for authors' contribution.

Author Response

Dear Reviewer,

We thank you again for your time and suggestions to improve our paper.

Julio Lyrio on behalf of the authors.

Reviewer 3 Report (Previous Reviewer 3)

Required modifications were carried out by the authors.

Author Response

Dear Reviewer,

We thank you again for your time and suggestions to improve our paper. 

Julio Lyrio on behalf of the authors.

Round 2

Reviewer 1 Report (Previous Reviewer 1)

The authors have indicated that they are no longer prepared to make any further changes to the manuscript, leaving me a choice between recommending either that it be accepted or be rejected. I have decided on the former, but the authors need to adopt a more constructive attitude in future.

This manuscript is a resubmission of an earlier submission. The following is a list of the peer review reports and author responses from that submission.

Round 1

Reviewer 1 Report

This manuscript describes the determination of the depth to basement of a sedimentary basin (with a constant density contrast) using underdetermined least-squares inversion. Information from seismic/borehole data is used to generate a starting model. In addition the depths obtained are constrained to lie within a priori limits by a logarithmic penalty function. This sort of inversion is common. Many papers have been published on the modelling of sedimentary basins in this manner, usually using densities that increase with depth (a geologically reasonable requirement) according to different functions eg linear, quadratic, etc. Since borehole data is available, was the density of the basin actually constant with depth ? (ie please plot Figure 12 with depths from 0-2200m).

The paper is very short and contains only a single synthetic data example and a single real data example. It would benefit from more of both. I suggest also using a 2D synthetic model initially because the results are easier to visualise, and of course the method is essentially identical.

Unfortunately there are no comparisons with other interpretation methods so the reader does not know whether the proposed method performs better than standard methods such as Li and Oldenburg etc. This needs to be done.

One factor of concern is edge-effects. Obviously the gravity anomalies at the edges of the data contain contributions from sources outside the survey area. Normally the dataset and the model are extended outside of the actual area of interest to account for this. In this case however the anomaly of interest only just fits within the limits of the dataset chosen, which is problematic. Indeed, edge-effects are clearly visible (eg see Figure 10b, particularly the NS and EW trending features in the SW and NE). Could the authors please comment ?

Could the authors please include information as to the computational cost of the proposed method, including aspects such as L-curve determination etc (ie runtimes, hardware, memory usage etc) ?

p.3; ‘This approach allows higher resolution..’; higher resolution than what ?

p.9; show the L-curve.

Figure 2; all axes need units.

Figure 3; show the forward response of the models and the difference between it and the (synthetic) observed data.

Figure 4; label the subplots a), b), etc and label all the axes.

Figure 5; display the data as images rather than filled contour maps, and add a colourbar (see also Figures 8a, 9b, 10a). State the grid interval in the caption.

Figure 6 is four plots but the caption only mentions a) and b). Make sure that they all have axes labels and units.

Figure 7 appears sunshaded but the caption does not say so. Please put the details in the caption and add a colourbar (and add units to the axes). See also Figures 8b, 9a, 10b

Arrange the references by first author surname rather than by number. Give specific page numbers when referencing books eg Blakely.

Reviewer 2 Report

The authors have presented a novel approach for estimating the relief of a surface that separates two media with varying densities. Their method involves the inversion of gravity data, integrating seismic interpretation, and borehole logs.

My main concern lies in the selection of relative densities for the sedimentary layers and the basin basement during the inversion process. There seems to be a significant disparity between the initial model and the final results (In the given field example, there is a maximum difference of 300 meters between the two, page 13, line 407). The impact of the relative density calculated based on the initial model on the results has not been thoroughly discussed in the paper, despite its crucial importance in gravity inversion.

In the model experimentation section, I did not come across any discussion regarding the initial model or the selection of densities. Additionally, there is no mention of how the well log data was incorporated as constraints. Furthermore, in Figure 3(c), there appear to be significant variations in the horizontal direction. Did the authors impose any smoothing constraints on the model? In my opinion, it is important for the authors to discuss how perturbations in the initial model and density selection can affect the inversion results.

Add a color bar to the contour maps in Figures 5 to 10. This will provide a clear reference for interpreting the values associated with the contour lines. Additionally, the numbers on the contour lines appear blurry and difficult to discern, so ensuring their legibility is important.

Reviewer 3 Report

The authors provided a good summary of inversion methodologies that stabilizes the undesired effects of the non-linearity. However all these are local optimizers in their nature. In my opinion it would be better to give a different idea to the readers in reconstructing the basement relief in a smoothed way, performed via global optimizers. There are some useful studies in the literature.

https://doi.org/10.1016/j.jappgeo.2015.03.008

https://doi.org/10.1016/j.jappgeo.2017.02.004

https://doi.org/10.1093/gji/ggaa492

https://doi.org/10.1093/gji/ggad222

Please add axes units such as meters in the maps of synthetic and real data cases.

In the real data case, the authors applied regional/residual separation methodology by using a linear trend. This is an important point because every different procedure changes the amplitude of the anomalies differently. This changes in amplitudes can affect the depth solutions obtained. For example, if we apply different order of polynomials to obtain regional effect, we will get the residual anomalies in different amplitudes, so different basement relief depths will be obtained. Therefore, how can we know the optimum. I think the authors must have made an assumption about this or used some prior knowledge. However, there is no information about this in the paper. I suggest the authors to explain this in detailed manner for the benefit of the readers.

Reviewer 4 Report

The manuscript describes a nonlinear inversion algorithm for recovering the basement surface in sedimentary basins.

The manuscript is very well written and the inversion technique is clearly described.

I have the following comments requiring some changes or some discussion in a revised version:

1) In the Introduction, is worth mentioning another method able to introduce prior information (borehole and seismic constraints) in the reconstruction of the morphology of a basement: Florio, G. 2018. “Mapping the Depth to Basement by Iterative Rescaling of Gravity or Magnetic Data.” Journal of Geophysical Research, [Solid Earth] 123 (10): 9101–20. https://doi.org/10.1029/2018JB015667. It is particularly interesting in that it does not require the assumption of a density contrast, which is a critical parameter in this kind of inversions, as even this manuscript demonstrates.

2) Formula (8): how the sensitivity matrix J is computed? please, insert a formula to illustrate how the elements of matrix J are calculated.

3) Page 10, lines 316-318: please, add a link where the reader can get the data.

4) Regional/residual separation: Discuss more deeply this fundamental step of the gravity modelling.

a) Show the regional gravity removed.

b) The regional should have been computed from an area much wider than that used for inversion (Figure 5a), so you may want to show in another figure the larger dataset used for this regional estimate.

c) How did you choose the degree of the used polynomial?

d) The two maps (5a and 5b) look very similar so that the removed linear trend shouldn't be much different from a constant…

5) What are the expected density contrasts between:

a) post-rift (shales) and pre-rift sediments (sandstones);

b) pre-rift sediments (sandstones) and basement ? (please describe the expected lithology of the basement)

I suspect that, given that the pre-rift sediments and basement densities are very similar, it would be more consistent to try to recover the top of pre-rift sediments by the inversion of the gravity data. Presently you constrain the model to fit the depth to the basement as observed in boreholes, but probably the interface originating the gravity anomalies lies at shallower depths, at the pre-/post-rift interface. A thorough discussion of the expected density contrasts is very important in the context of this manuscript.

6) Another problem that was not discussed in the manuscript is the presence of horizontal density contrasts in the sedimentary cover, as it results from the section in figure 4. In fact, pre- and post-rift sediments may have a substantial density contrast.

7) Related to the previous point: it should be stated that, similarly to other algorithms used to find the depth to basement, even in this case it is assumed that the modelled residual gravity field is due solely by basement undulations and that the sedimentary cover has no horizontal density contrasts (horizontal layering).

8) page 13, lines 366-367: actually, application of the stripping technique may not be so easy: you need a very detailed and accurate set of constraints, otherwise this kind of residuation does more harm than good.

9) page 13, lines 375-376: looking at Figure 12, it appears that the shales have a density of about 2.4 g/cm3, and the sandstones and basement have a density of about 2.65. It results that the assumed density contrast of 0.39 g/cm3 seems to be in disagreement with the density logs, which would suggest a contrast of about 0.25 g/cm3.

10) Conclusions, line 466: I am not sure you demonstrated that a constant density is appropriate in the studied case.

11) All the maps: as the small labels on the contours are not well readable, please add a colorbar.

Round 2

Reviewer 1 Report

Unfortunately the authors have ignored almost all of the necessary corrections, even some of the minor ones such as corrections to the references or the correct labelling of figures.

Previously I asked ‘Since borehole data is available, was the density of the basin actually constant with depth ? (ie please plot Figure 12 with depths from 0-2200m).’, but the authors did not respond. I understand that the proposed approach can be used in situations where there is no borehole information, but in the particular real data case that is presented in this manuscript, was the density constant with depth ??

As pointed out previously, the paper is very short and contains only a single synthetic data example and a single real data example. It would benefit from more of both. I suggest also using a 2D synthetic model initially because the results are easier to visualize, and of course the method is essentially identical.

As commented previously, unfortunately there are no comparisons with other interpretation methods so the reader does not know whether the proposed method performs better than standard methods such as Li and Oldenburg etc. This needs to be done. The authors response is that they they are presenting an approach that works and that that is sufficient. It is not, particularly if the proposed approach has significant disadvantages compared to existing methods (eg it is computationally very slow, or it is less accurate, etc).

As commented previously, ‘Figure 3; show the forward response of the models and the difference between it and the (synthetic) observed data.’. I understand that the model is being evaluated only at certain points, but if the model response does not even fit the data then the model will not be valid.

Previously I stated ‘p.3; ‘This approach allows higher resolution..’; higher resolution than what ?’. Unfortunately the authors have not clarified what they mean in the text of the manuscript.

Previously I requested ‘Could the authors please include information as to the computational cost of the proposed method, including aspects such as L-curve determination etc (ie runtimes, hardware, memory usage etc) ?’ and they authors respond that this is not relevant. I disagree. There is little point introducing a new strategy if it is unusable in practice.

Previously I requested ‘p.9; show the L-curve’; the authors response is that their code does not display the L curve, it just calculates it in the background and picks the optimum value automatically. Relying on software to automatically choose any important model parameters without any possible method of verification is an extremely bad way to do anything, not just geophysics. Since the authors have the original code I strongly suggest that they write the L-curve values to disk and then plot them to check on the software.

Figure 5; as requested previously, display the data as an image, not a contour map with a large contour interval. See also Figures 7,8b, 9a, 10b.

Figure 5; as requested previously, state the grid interval in the caption (not just somewhere in the manuscript text).

As pointed out previously, Figure 6 is four plots but the caption only mentions a) and b). Make sure that they all have axes labels and units. See also Figure 4. Problems with the ‘automatic label’ software are not relevant. Please mark the interpreted basin depths on the seismic section (preferably in a larger plot) before and after the inversion of the gravity data.

As requested previously, give specific page numbers when referencing books, eg Blakely. It is not reasonable to expect a reader to to search an entire book for a particular equation.

Reviewer 4 Report

Unfortunately, Authors did not exploit this revision round as an occasion to improve their manuscript.

Authors provided a very quick response. This was only possible because they do not want to edit their manuscript.  I can see that my comments have not been satisfactorily answered or not answered at all.

In particular, I am disappointed by the Author's response to the following points:

4) Regional/residual separation; 5) discussion about the actual surface responsible for the main gravity signal; 9) density logs vs. density contrast used.

These are important issues, and if the authors had taken them seriously the research would have become more credible and the manuscript would have had greater impact.
I don't think it's helpful for me to review the manuscript again.
